# Cellular Composition and Differentiation Signaling in Chicken Small Intestinal Epithelium

**DOI:** 10.3390/ani9110870

**Published:** 2019-10-27

**Authors:** Haihan Zhang, Dongfeng Li, Lingbin Liu, Ling Xu, Mo Zhu, Xi He, Yang Liu

**Affiliations:** 1Department of Animal Sciences, Hunan Agricultural University, Changsha 410128, Hunan, China; hhzhang@vt.edu (H.Z.); xulingous@gmail.com (L.X.); 2Department of Animal Genetics, Breeding and Reproduction, College of Animal Science and Technology, National Experimental Teaching Demonstration Center of Animal Science, Nanjing Agricultural University, Nanjing 210095, China; lidongfeng@njau.edu.cn (D.L.); 2018105017@njau.edu.cn (M.Z.); 3Medical Sciences, Indiana University School of Medicine, Bloomington, Indiana, IN 47408, USA; 4College of Animal Science and Technology, Southwest University, Chongqing 400715, China; liulb515@163.com

**Keywords:** differentiation, chicken, small intestine, epithelium

## Abstract

**Simple Summary:**

The small intestine is the major place for chickens to digest and absorb the nutrients. The intestinal epithelium is a single layer of cells that are generated by the stem cells in the crypt. This review mainly summarized the recent findings of molecular signaling pathways that control the proliferation and differentiation of chicken small intestinal stem cells, and the biological functions of chicken small intestinal stem cells, and the biological functions of chicken small intestinal epithelium corresponding to different cell types. We also provided some novel in vitro models for studying chicken small intestinal stem cells. With the understanding of cellular composition and differentiation in chicken small intestinal epithelium, it may bring the new ideas for the poultry industry to improve the small intestinal health and nutrition.

**Abstract:**

The small intestine plays an important role for animals to digest and absorb nutrients. The epithelial lining of the intestine develops from the embryonic endoderm of the embryo. The mature intestinal epithelium is composed of different types of functional epithelial cells that are derived from stem cells, which are located in the crypts. Chickens have been widely used as an animal model for researching vertebrate embryonic development. However, little is known about the molecular basis of development and differentiation within the chicken small intestinal epithelium. This review introduces processes of development and growth in the chicken gut, and compares the cellular characteristics and signaling pathways between chicken and mammals, including Notch and Wnt signaling that control the differentiation in the small intestinal epithelium. There is evidence that the chicken intestinal epithelium has a distinct cellular architecture and proliferation zone compared to mammals. The establishment of an in vitro cell culture model for chickens will provide a novel tool to explore molecular regulation of the chicken intestinal development and differentiation.

## 1. Introduction

The small intestine is the major site for nutrient digestion and absorption in the domestic chicken (Gallus gallus). The small intestinal epithelium contains nutrient transporters and digestive enzymes, secreting hormones and functioning as a physiological barrier against pathogens by producing glycoproteins and defensins [1,2]. An internal folding of small intestinal epithelium forms intestinal crypts and an external protruding generates intestinal villi, which are directly exposed to the luminal environment [3]. Stem cells reside in the crypt, and the crypt niche provides essential signals and regulators for their proliferation and differentiation. Stem cells proliferate and differentiate along the crypt-villus axis and give rise to mature, terminally differentiated cells that are distributed along the intestinal villi. These cells play the roles of absorption and secretion [4,5]. In chicken, proliferating cells are also found along the intestinal villi, possibly due to cell production dynamics that differ from humans [6]. Additionally, the presence of Paneth cells in the intestinal crypts varies in distinct avian species [7,8,9]. Even though knowledge of chicken intestinal epithelial characteristics is limited, recent research has found that the chicken small intestinal epithelium shares some similarities, although it significantly differs from human and mice and has unique characteristics. Thus, it is critical to understand the detailed characteristics of the chicken small intestinal epithelium. In this review, we summarized recent important findings involved in the development and differentiation of the small intestinal epithelium in chickens, compared with human and mice studies, in order to outline the preliminary physiological structure and bring forth novel ideas for research on chicken small intestinal epithelium.

## 2. The Development of Chicken Small Intestine

The chicken small intestine is formed by the gut tube, which is derived from the internal folds from anterior and caudal intestinal portals of the embryonic endoderm [10]. Gut tube formation is completed and regulated by the complex signaling that is induced by either Sonic hedgehog or Wnt [11,12], at the time when the umbilicus is only slightly open for the yolk stalk that connects to the yolk sac around embryonic day e5 [13]. Between e5 and day of hatch, the small intestine elongates, but the epithelium can be observed only at the end of the embryogenesis [14]. Peristalsis within the chicken small intestine was also detected at e5, which suggests that mechanical intestinal functions arise during early embryonic development [15]. In contrast, expression of nutrient transporters, mineral transporters and digestive enzymes was detected only during the last stages of embryonic development [2,16,17,18]. Physiological structures, including intestinal villi and crypts, were fully established by day of hatch [16]. Mature and functional epithelial cells of the chicken small intestine, such as goblet cells and enterocytes, have been histologically detected by e18 [19]. Since chickens are precocial species and their intestines are fully matured at the end of the embryogenesis, they are capable of independent feeding immediately after hatch. Therefore, formation of crypt-villi structures and a functional epithelium is prerequisite before hatch.

## 3. Cellular Proliferation in the Chicken Intestinal Epithelium

The intestinal crypt, which was first identified by Jonathan Lieberkühn, is the site of stem cells with proliferating abilities for self-renewal and differentiation, thus maintaining homeostasis in the intestinal epithelium. In mammals, the stemness and differentiation of intestinal stem cells are controlled by complex cell signaling that is mainly modulated by cell–cell communication from Paneth cells within the crypts or stromal cells surrounding the crypts [4,5,20]. Differentiation of intestinal stem cells initially forms transit amplifying (TA) proliferating cells which rapidly propagate the epithelial cell population and migrate either out the intestinal crypt to generate the terminally-differentiated cells along the villus (enterocytes, goblet cells, enteroendocrine cells) or deep into the crypts for differentiation into Paneth cells (Figure 1A) [21]. Thus, the proliferating zone in mammalian intestinal epithelial cells is located in the crypts. 

However, in chicken, proliferation of epithelial cells is not restricted to crypts, but is also present along the villus (Table 1). This was found by measuring the uptake rate of ^3^H-thymidine. Additionally, immunostaining of 5-bromo-2-deoxyuridine (BrdU) and proliferating cell nuclear antigen (PCNA) in the chicken small intestine was also located along the villi. However, the majority of proliferating activities was restricted to the crypts (80%) [6,22]. Bar-Shira and Friedman also detected a few proliferating columnar cells with condensed chromosomes that were located in the chicken duodenal villus by hematoxylin and eosin staining. They also found that the lysozyme-secreting cells, potentially chicken Paneth cells, were distributed in both crypts and villi. Due to the important effects of Paneth cells on stem cell proliferation and differentiation, one hypothesis is that the proliferating cells identified in chicken intestinal villi were either intestinal stem cells or TA proliferating cells [23]. However, in a study conducted by our group, we have localized cells expressing *Lgr5* (Leucine-enriched G-protein-coupled receptor 5) and *Olfm4* (Olfactomedin 4) genes, which are the two widely-used intestinal stem cell makers in the chicken small intestinal crypts [24]. Thus, chicken intestinal stem cells are potentially restricted to the crypts, and the proliferating cells that were detected along the villi were probably delayed-differentiated TA cells. Furthermore, the partially differentiated TA cells were also potentially identified at the zone above the Olfm4 positive stem cells in the crypts and beneath the *PepT1* (Oligo-peptide transporter 1) positive enterocytes along the villus [24,25]. 

## 4. Notch and Wnt Signaling Cooperate for the Intestine Epithelial Differentiation

In the mammalian model, differentiation of intestinal stem cells is mainly mediated by the interactions with neighboring cells or the microenvironment, which provide important factors that activate two major signaling pathways, Notch and Wnt. These pathways synergistically control the direction of intestinal stem cell differentiation into either an absorptive cell linage (enterocytes) or a secretory cell linage (goblet cells, Paneth cells, and enteroendocrine cells) (Figure 2) [21,33,34].

In chickens, the mechanism of Notch signaling in mediating differentiation of intestinal stem cells has not yet been examined. Notch signaling in mice is a critical pathway for controlling the fate of the intestinal stem cells. The activation of the Notch pathway inhibits the formation of secretory cells such as goblet cells, enteroendocrince cells, and Paneth cells, from intestinal stem cells, whereas the suppression of Notch signaling results in increased amounts of secretory cells due to the decrease of differentiation from stem cell to absorptive cells (Figure 1B) [35,36,37]. In the chicken small intestine, melatonin was found to promote the augmentation of the secretory goblet cell population. Meanwhile, levels of mRNA expression of key Notch receptors (Notch1 and Notch2) and ligands (Dll1 and Dll4) that control intestinal stem cell differentiation were significantly reduced in melatonin-treated birds [38]. This suggests that differentiation of intestinal stem cells into a secretory cell linage is possibly promoted by the inhibition of Notch signaling pathway in chicken epithelium. In addition, the transcriptional expression of Olfm4, which is an important Notch target gene, was vastly elevated in chicken intestinal stem cells [24]. Notch signals control intestinal stem cell differentiation into a secretory cell linage also through the transcriptional regulation of *Hes1* (Hes family bHLH transcription factor 1) (Figure 2), which was found as a direct inhibitor of *Atoh1* (Atonal bHLH transcription factor 1) expression and causes the repression of intestinal stem cell differentiation into secretory cells (Figure 1B) [39,40]. The chicken c-hairy-1, which is a homologue of Hes1 and is regulated by Notch signaling, was found to co-express with Musashi-1, a potential intestinal stem cell maker that activates Notch signaling by enhancing Hes1 expression. These proteins were located in the chicken small intestinal crypts, where intestinal stem cells are located [41]. Taken together, the activation of Notch signals is important for self-renewal and differentiation of intestinal stem cells in chicken. However, recently, Bar Shira and Friedmen utilized lysozyme as a marker for identifying Paneth cells and found that lysozyme-secreting cells were not restricted to the crypts but were located along the intestinal villus [21]. Paneth cells are a critical cell population that provide Notch ligands, including Dll1 and Dll4, for maintaining differentiation and self-renewal abilities of intestinal stem cells. The loss of Paneth cells in chicken intestinal crypts might result in the ablation of Notch ligand, causing inhibition of Notch signaling. However, mesenchymal cells and other mature epithelial cells that are adjacent to the intestinal stem cells in the crypts also provide important signaling for the Notch ligand [42].

The canonical beta-catenin/Wnt signaling is also a critical pathway for determining differentiating intestinal epithelial cell fates through multiple points of crosstalk between Notch signaling (Figure 1B). *Math1* or *Atoh1* is an important transcription factor which is negatively regulated by Notch signaling (Figure 2) [44]. Increased expression of Atoh1 has been shown to induce intestinal progenitor cell differentiation into secretory cells [45]. Some studies found that upregulation of Wnt signaling activates Atoh1 expression and differentiation of intestinal stem cells into secretory cells [46,47]. Lgr5 is a leucine-enriched G-protein coupled receptor that activates the canonical Wnt signaling pathway through binding R-spondin 1–4, and it is widely used as a biomarker for identifying intestinal stem cells [26]. Localization of Lgr5 mRNA expression in the chicken small intestine was restricted to stem cells in the crypts and no significant differences of mRNA expression were found between different segments of the small intestine [24]. However, the express of Lgr5 ligand R-spondin1 was higher in jejunum and ileum than in the duodenum in chickens [27]. This may indicate that hyper-activation of Wnt signaling occurs in the chicken jejunum and ileum. However, localization of R-spondin1 within the intestinal epithelium and its downstream effects still need to be investigated. The canonical Wnt signaling pathway is initiated by the binding of Wnt proteins to the Frizzled receptors (Figure 2). In chicken, the distribution of Wnt and Frizzled proteins within the intestine during early embryonic development was visualized by whole-mounted in situ hybridization analysis. Wnt5a was identified as the most prevalent Wnt protein within the chicken small intestinal epithelium from embryonic day 4 to 8. Frizzled 7 was the only receptor that was expressed by the small intestinal endoderm, which specializes as the mucosal layer of the small intestine later on throughout embryonic development [14]. However, tissue distribution and the cellular localization of Wnt5a and Frizzle 7 proteins in chicken small intestine during the posthatch stage have not yet been identified. The Wnt and Notch signals synergistically regulate intestinal stem cells to differentiate into either absorptive cell lineages, including enterocytes, or secretory cell lineages, including goblet cells, enteroendocrine cells, and Paneth cells.

## 5. Cell Types along the Intestinal Epithelium

### 5.1. Absorptive Enterocytes

Enterocytes are the most abundant intestinal epithelial cell population (>80%), and mainly participate in nutrient digestion and absorption [28]. These polarized cells along the intestinal villi are connected to each other through tight junctions and form a physiological barrier that can distinguish antigens from digestible nutrients. The surface of the enterocyte apical membrane is vastly increased by microvilli, which contain digestive enzymes and nutrient transporters and create the functional intestinal brush border [48]. In chicken, mature, columnar-shaped enterocytes with apical microvilli and lateral tight junctions are present at day of hatch [49]. The main function of the enterocytes is the uptake and transfer of nutrients, which is conducted by transporters located in the apical brush border and basolateral membranes. Nutritional substrates are metabolized into small molecules by digestive enzymes, absorbed by brush border membrane transporters and metabolized into smaller molecules. They are then resynthesized into other compounds for enterocyte utilization, or transported into the blood through the basolateral membrane transporters [31,50]. Chicken small intestinal enterocytes were found to exclusively express mRNA of PepT1, an oligo-peptide transporter, from the top of the intestinal crypts to the tip of the villi [25]. However, the expression of *SGLT1* (Sodium-dependent glucose co-transporter 1) mRNA in the chicken small intestinal epithelium was not only detected along the villus, but also located within the crypts, suggesting that glucose intake occurs within chicken intestinal stem cells [51].

### 5.2. Goblet Cells

An important secretory cell type along the crypt–villus axis that maintains intestinal homeostasis is the goblet cell. The major function of goblet cells is to secrete mucins and other products, which together form the mucus layer, an innate immune defense barrier, to protect the intestinal epithelium. This cell population’s name is linked to its specialized morphology, which is goblet shaped and has distended theca that accumulates with highly glycosylated granules in proximity to the basal membrane [52]. In mammals, goblet cells are derived from intestinal stem cells, and this process has been shown to be mainly controlled by Notch signaling [53]. The suppression of Notch signaling promotes stem cell differentiation into secretory cells, which are goblet cells, Paneth cells, and enteroendocrine cells, due to the inhibition of stem cells differentiated into absorptive cells [37]. In chicken, goblet cells were early present around embryonic day 17, and only contained acidic mucins. The percentage of mature goblet cells in chicken are gradually increased from jejunum (23%) to ileum (26%), and the morphology of the goblet cells are well-formed and enclosed with both acidic and neutral mucins during the posthatch stage [19]. Additionally, recent findings indicated that melatonin may inhibit Notch signaling, which can cause the increase of the small intestinal goblet cells in the chicken [38]. Aside from histological staining such as PAS (Periodic acid Schiff-base) and Alcian blue, the *Muc2* gene is an goblet cell marker [54]. Thus, the change of Muc2 expression is commonly used as a means to detect the changes in quantity functionality of intestinal goblet cells. Li et al. verified the effect of melatonin on chicken goblet cell differentiation by quantitating the expression of Muc2 in the small intestine [38].

### 5.3. Paneth Cells

Paneth cells were first discovered in 1888. Their location along the intestinal epithelium is dependent on the species, but usually identified in the crypts [9]. Paneth cells are characterized by their trapezoidal shape and cytoplasmic granules located at the base of the cell [55]. The Paneth cell can be identified by histological analyses including phloxine-tartrazine and eosin staining, because it contains acidophilic granules, which are concentrated with cationic charged proteins or peptides [55]. Paneth cells appear in both the small intestine and colon in humans at the beginning of the second trimester of pregnancy [56]. In mice and rats, Paneth cells do not appear until the intestinal crypts are completely formed, after birth [29]. In chicken, the presence of Paneth cells in the intestine is controversial. An immunohistochemical analysis of the lysozyme protein, which is a widely used biomarker for labelling Paneth cells, showed that lysozyme was distributed within cells along the villus epithelium of 17 day old chickens, but not present in the intestinal crypts [30]. However, in another study, Paneth cells were found to be present in the small intestinal crypts of six-month-old chickens when treated with histological staining and in situ hybridization of lysozyme c [32]. In other avian species such as ostrich, Paneth cells were found to be absent [9]. Some other animals such as *Xenopus* were found to not have Paneth cells in the crypts, but had other cell types along the villus that function like Paneth cells [57].

### 5.4. Enteroendocrine Cells

The gastrointestinal tract regulates gut–brain and brain–gut neuroendocrine bidirectional interactions. Enteroendocrine cells, even though they are sparse compared to other intestinal epithelial cell populations (about 1%), play a very important role for animals in regulating hormone secretion, gastrointestinal enzymatic activity, and feeding behavior [58]. There are different types of enteroendocrine cells that are named with letters according to the major secretory products they contain. Cell types that are located in the small intestine include I, K, S, and D cells [59]. I cell is an enteroendocrine cell that secretes cholecystokinin (CCK), which controls the secretion of bile and pancreatic enzymes, and regulates satiety. Cells that are able to produce gastric inhibitory peptide (GIP) are called K cells. S and D cells are cells that secrete secretin and somatostatin, respectively [59,60]. Glucagon-like peptide-1 (GLP-1), which functions to boost the secretion of insulin and attenuate the production of glucagon, is a popular biomarker for identifying L cells in animal intestines [61]. In chicken, L cells are primarily detected in the intestinal epithelium in the jejunum and ileum, but barely found in the duodenum [62]. GLP-1 positive cells were distributed from the middle of the intestinal villi to bottom of the crypts, and the maturity of the granule enteroendocrine L cells were supposed to occur in the crypts [63,64,65]. Immunoelectron microscopy is a method for visualizing cellular ultra-structures that express a protein of interest. Watanabe et al. applied this technique to demonstrate that chicken GLP-1 positive cells were flask-shaped and abundant with secretory granules that enclosed the GLP-1 proteins [61]. In chicken, the number of GLP-1 positive endocrine cells in the ileum was found to be associated with the intake of dietary proteins and amino acids [66,67].

## 6. In Vitro Approaches for Studying Chicken Intestinal Epithelium

Many immortal cell lines of the mammalian intestinal epithelium are available to obtain. However, they are artificially modified and not capable of reflecting the true dynamics of intestinal epithelial cells as well as primary cells. Currently, there are no commercial intestinal epithelial cell lines available for chicken. However, several studies have successfully established methods for isolation and culture of chicken primary intestinal epithelial cells (IEC). The primary IEC were isolated from chicken either at late embryonic stage (e15–e18) or day 1 post-hatch [68,69,70], but some were obtained from chickens older than 6 weeks [71,72,73]. The isolation procedure and culture medium for chicken primary IEC are similar in each study, whereas Kaiser et al. displayed a detailed cell isolation protocol and comprehensive characterization of tissue morphology, cadherin and cytokeratin expression, and cellular ultra-structures of chicken primary IEC [71]. A chicken primary IEC culture would provide a novel way to investigate physiology and differentiation of cells lining the intestinal crypt–villus axis, along with their interaction with nutrient supplements or pathogens. However, primary cells have a short life span, typically 5–6 days in vitro. After that, the cells lose contact with the stroma cells or submucosal cells. Thus, chicken intestinal crypt cells were found to deplete the ability to continuously proliferate in vitro. The newly-developed three-dimensional organoid originated from intestinal crypts retains the essential architecture for stem cell differentiation and cellular renewal and allows a long-term viable culture, and this new technique would provide a great mimic model of the intestinal crypt niche because it not only generates the culture of the intestinal stem cells, but also cultures the Paneth cells or stromal cells that offer the important cellular signals for stem cell differentiation and proliferation [74,75,76]. Li et al. have extrapolated the organoid 3D culture for chicken small intestinal crypts and have successfully established a Matrigel system for culturing chicken intestinal crypts, which nicely formed the protruding villus structure and expressed important intestinal stem cell maker genes. [77]. In addition, Pierzchalska et al. found that prostaglandin E2 was important to support the culture of chicken intestinal organoid that was isolated from embryonic tissue in Matrigel matrix [78]. Another way to generate 3D culture for chicken intestinal organoid was created by Panek et al. by using a hanging drop culture, which differs to Matrigel culture and removes the solid artificial scaffold [79]. Another elegant in vitro model for studying chicken enterocytes was conducted by Rath et al. [80]. They developed a method for separating and culturing primary chicken enterocytes, which harbored alkaline phosphatase activity and expressed cadherin. They provided an excellent ex vivo model to investigate nutrient absorption and immune regulation of enterocytes, particularly in chicken small intestine.

## 7. Conclusions

The multifunctional intestinal epithelium has a heterogeneous and organized cellular architecture. This single-layer epithelium absorbs nutrients, secretes digestive enzymes, and functions as a protective barrier against pathogens. Chickens are precocial animals; therefore, their intestine fully matures during the late embryonic stage. Various fundamental studies from the past decades have examined chicken embryonic intestinal development. However, detailed cellular configuration and regulation of differentiation in the chicken intestinal epithelium is still poorly understood. In this review, current knowledge regarding chicken intestinal epithelium dynamics and physiology was detailed by introducing gut developmental processes, differentiation, regulation, and cellular dynamics. Furthermore, similarities and differences between chickens and humans or mice were discussed. Lastly, we briefly presented recent in vitro models for examining the chicken small intestinal epithelium, particularly in order to provide novel strategies or ideas for future studies on molecular signaling that control cellular proliferation and differentiation in chicken intestinal stem cells. This review provides a base for further applicable studies on the effects of pathogenic infections and nutritional supplementations on differentiation or development of the chicken small intestinal epithelium. On the basis of the previous literature, chicken was found to be different from the mammals and rodents in terms of (1) the fact that proliferation of intestinal epithelial cells occurs from crypt to villus, and (2) the presence of Paneth cells in the small intestine is controversial, probably presenting in both the crypt and villus. These findings may indicate that the differentiation and proliferation of chicken intestinal epithelium have a distinct underlying mechanism. Thus, it is important for the further investigation to understand the unique knowledge for chicken intestinal epithelium so that we can accurately analyze the nutritional or microbial interactions that affect the intestinal epithelial homeostasis in chicken.

## Figures and Tables

**Figure 1 animals-09-00870-f001:**
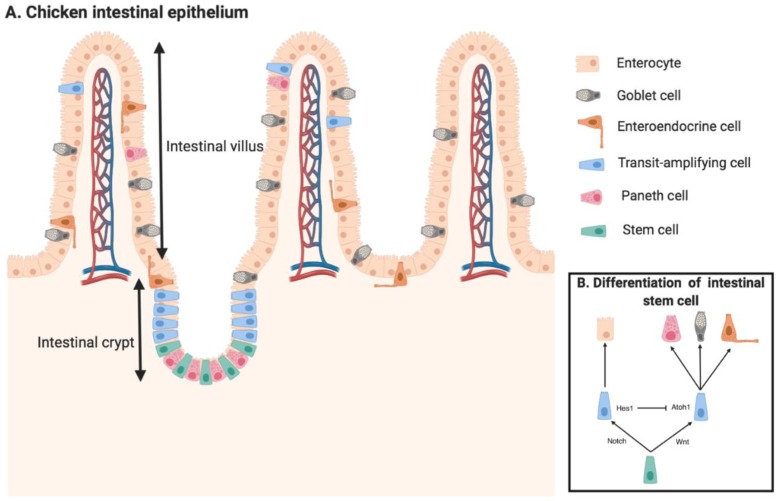
Chicken small intestinal epithelium model. The chicken intestinal epithelium is a complex biosystem that has multiple functions. (**A**) The intestinal crypt invaginates as a pocket and the intestinal villus protrudes as a finger structure. Intestinal stem cells are located at the crypts and differentiate into transit-amplifying cells for further differentiation into mature epithelial cell populations (enterocytes, goblet cells, enteroendocrine cells, and Paneth cells). (**B**) The model for the differentiation of the intestinal stem cell is a process of the intermediate differentiation from stem cells to transit-amplifying (TA) cells, and the terminal differentiation from TA cells to absorptive enterocytes, secretory goblet, Paneth, and enteroendocrine cells, which is mediated by the synergistical regulation of Notch and Wnt signals.

**Figure 2 animals-09-00870-f002:**
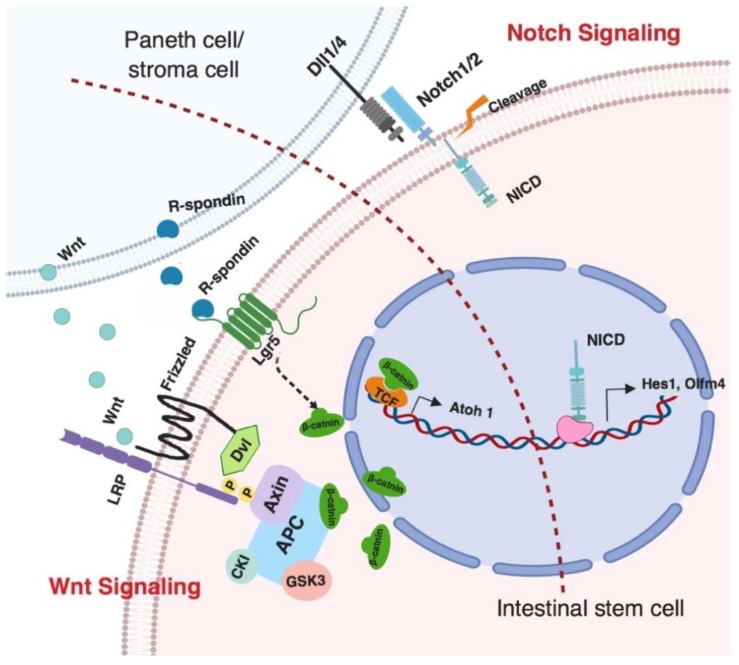
Notch and Wnt signaling pathways. The differentiation of intestinal stem cells is mainly controlled by Notch and Wnt signaling pathways. The neighboring cells, including Paneth or stromal cells, provide Wnt proteins and R-spondins for activating Wnt signaling, and Notch ligands Dll1/4 for inhibiting Notch signaling. The canonical Wnt signaling is activated by the binding of Wnt protein binding to the *LRP* (Low density lipoprotein receptor-related protein)/Frizzled receptor complex to initiate the dissociation of intracellular *APC* complex and cause the release of β-catenin, which binds the transcriptional factor *TCF* and activates the expression of *Atoh1* that controls intestinal stem cell differentiation. *Lgr5* is an intestinal stem cell marker and an R-spondin ligand. The activation of Lgr5 also promotes the accumulation of β-catenin and causes the downstream activation [43]. Notch signaling is controlled by interactions of Notch and Dll1/4. The dissociation causes the cleavage of Notch receptor to release the NICD (notch intra-cellular domain), which is able to interact with other transcriptional factors to initiate the expression of *Hes1* or *Olfm4* for regulating stem cell differentiation in intestine [34].

**Table 1 animals-09-00870-t001:** Similarities and differences of chicken small intestinal epithelium compared with mammals.

Cell Types	Similarities	Differences	
		Chicken	Mammal
Enterocytes	Polarized columnar cells along the villus with nutrient absorptive ability [26,27,28]		
Paneth cells		Whether presented in the crypts is still controversial; could be distributed along the villus, or in the crypt [29,30]	Acidophilic granulocytes that are located in the crypt and interspersed with stem cells [19,24]
Goblet cells	Mucus-secreted cells that are distributed along the villus and in the crypt [31]		
Stem cells		Located in the crypt but not interspersed with Paneth cells [22]	Distributed in the crypt with Paneth cells inserted [19,24]
Enteroendocrine cells	Minor cell population along the crypt-villus axis, secreting important hormones related to digestibility and feed intake [32]		
Proliferation zone		Intensively identified in the crypt, but still detected along the villus [6,20]	Constraint in the crypt

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
