# Peer review of "Cellular Composition and Differentiation Signaling in Chicken Small Intestinal Epithelium"

_animals, 2019, doi:10.3390/ani9110870_

Round 1

Reviewer 1 Report

No further comments. Excellent review.

Author Response

Dear Editor,

Thank you for your time to go through the manuscript again. And thank you for all the compliments for the article.

Sincerely,

Haihan

Reviewer 2 Report

Reading the revised version of the text I have noticed the following problems:

I. Although the authors introduced the most of suggested changes to the body text and figures the text still needs some editing.

For example lines listed below contain errors or are difficult to understand:

line 2. "Cellular construction and differentiation signaling in chicken small intestinal epithelium" - I would suggest the term "cellular composition".

line 195-6. Paneth cells were first discovered in 1888. They are located in the crypts but present dependent on the species.

line 224-225 GLP-1 positive cells were detected to distribute from the middle of the intestinal villi to the crypts.

line 247 The newly-developed three-dimensional organoid culture for intestinal crypts - should be "originated from intestinal crypts"

line 248-252 "stem cell population for differentiation" - it is difficult to follow what this very long sentence means?

II. The table 1 and Figure 1 are now inconsistent (the location of Paneth cells).

Author Response

Dear Editor,

Thank you for all helpful comments. We have revised our manuscript according to your suggestions. Please check the attachment for the new version.

Thank you!

Sincerely,

Haihan

This manuscript is a resubmission of an earlier submission. The following is a list of the peer review reports and author responses from that submission.

Round 1

Reviewer 1 Report

In this review article entitled " Development and differentiation of chicken small intestinal epithelium” by Haihan Zhang and co-authors presented their review on different cell types and pathways of differentiation of small intestinal epithelium in chicken.  The present review focused on the current knowledge regarding chicken intestinal epithelium dynamics and physiology by introducing gut developmental processes, differentiation, regulation and cellular dynamics. Furthermore, similarities and differences between chickens and humans or mice were discussed in detail.

The presented data are sound. However, there are a few points that need to be considered.

Authors are suggested to strengthen the importance and relevance of the review and future directions in the manuscript. References should be cited by following journal style/format. Need to check for typographical errors, plagiarism, punctuation, and grammar throughout the manuscript.

Author Response

Dear Reviewer:

Thank you for all the comments. I have revised the manuscript according to the comments and highlights all the revisions in green that correspond to your comments. We enclosed the revised revision here.

We have double-checked the format of all the references. They should abide by the format of the journal.

Thank you!

Sincerely,

Haihan

Reviewer 2 Report

Please see more detailed information in the attachment.

Author Response

Dear Reviewer,

Thank you for all the comments. We totally accept all the suggestions. We have accepted all the minor grammar changes in the manuscripts. For the comments, We replied in the main text and highlighted them in yellow. Please see our revisions from the attachment.

Additionally, for the questions related to the figure and the captions, we change the figure, and listed the unique characteristics of chicken intestinal epithelium in a table which is Table. 1. 

Thank you for your comments!

Sincerely,

Haihan

Reviewer 3 Report

The paper is generally interesting and written in clear and logical form. But some parts of it needs improvements. In my opinion the major shortcomings of the review are:

The part untitled The development of chicken small intestine (L51-63) is very short. It lacks the description of avian gut and comparison between chicken and mammalian small intestine development. The subject is stressed in the title so it should be longer or the title should be modified (There are many papers which can be cited in this part e.g. D.J. Roberts, R.L. Johnson, A.C. Burke, C.E. Nelson, B.A. Morgan, C. Tabin, Development 1995 121: 3163-3174) The part untitled In vitro approaches for studying chicken intestinal epithelium (L229-L253) As far as organoids culture of avian origin are concern some important papers are missing (e.g. 1. Yue-Bang Yin et s. J. Agric. Food Chem (2019), 67, 2421−2428;, 2. Robin H. Powell and Michael S. Behnke Biology Open (2017) 6698-705 doi:10.1242/bio.02171; 3. Pierzchalska, M. et al. (2012). BioTechniques 52, 307-315; 4. Panek, et al. Cytotechnology (2018), 70, 1085−1095). There are truly only few papers published on chicken intestinal epithelial cell culture models so it would be better and possible to cite all of them. The figures are not very original as similar drawing can be found in other reviews on mammalian gut epithelium. It will be beneficial to add a figure or a table providing the differences and similarities between avian and mammalian systems

Author Response

Dear Reviewer,

Thank you for all the comments, we totally accept all your comments. We tried our best to cite more references including the references you listed in your comments. We replied and highlighted our revisions in Blue. We also added a new table (Table 1) that summarizes the similarities and differences of intestinal epithelium between chicken and mammals. We enclosed our new version here, please see the attachment.

Thank you!

Sincerely,

Haihan
